# Molecular Mechanisms and the Significance of Synonymous Mutations

**DOI:** 10.3390/biom14010132

**Published:** 2024-01-20

**Authors:** Peter Oelschlaeger

**Affiliations:** Department of Biotechnology and Pharmaceutical Sciences, College of Pharmacy, Western University of Health Sciences, Pomona, CA 91766, USA; poelschlaeger@westernu.edu

**Keywords:** synonymous mutation, silent mutation, codon usage bias, mRNA secondary structure, translation efficiency, genetic code, degeneracy

## Abstract

Synonymous mutations result from the degeneracy of the genetic code. Most amino acids are encoded by two or more codons, and mutations that change a codon to another synonymous codon do not change the amino acid in the gene product. Historically, such mutations have been considered silent because they were assumed to have no to very little impact. However, research in the last few decades has produced several examples where synonymous mutations play important roles. These include optimizing expression by enhancing translation initiation and accelerating or decelerating translation elongation via codon usage and mRNA secondary structures, stabilizing mRNA molecules and preventing their breakdown before translation, and faulty protein folding or increased degradation due to enhanced ubiquitination and suboptimal secretion of proteins into the appropriate cell compartments. Some consequences of synonymous mutations, such as mRNA stability, can lead to different outcomes in prokaryotes and eukaryotes. Despite these examples, the significance of synonymous mutations in evolution and in causing disease in comparison to nonsynonymous mutations that do change amino acid residues in proteins remains controversial. Whether the molecular mechanisms described by which synonymous mutations affect organisms can be generalized remains poorly understood and warrants future research in this area.

## 1. Introduction

It has been eight decades since DNA was recognized as the carrier of genetic information by Avery and coworkers [1] and seven since the structure of the DNA double helix was solved by Franklin and Gosling [2], Wilkins and coworkers [3], and Watson and Crick [4,5]. Based on the fact that, according to their model, adenine (A) always pairs with thymine (T), and guanine (G) always pairs with cytosine (C), Watson and Crick proposed that any sequence of bases is possible on one strand, but that it determines the sequence of the bases on the other strand, which runs in the opposite direction [4]. Such variability of a four-letter code and complementarity of the two strands provided the basis of the genetic code and a mechanism of precise replication. They hypothesized that in order to replicate the two DNA strands, they would have to be unwound and separated, and complementary nucleotides would have to be placed according to what is now called Watson–Crick base pairing to form a new, complementary DNA strand that would combine with the original template to form a new DNA double helix [4].

In the following years, several hypotheses were formulated [6]. The Sequence Hypothesis states that the sequence of nucleotides in DNA determines the sequence of amino acids in proteins. The Central Dogma states that information flows in one direction only, from genes to proteins. The Adaptor Hypothesis proposes that there are adaptor molecules (now known as transfer or tRNAs) that base pair with the “template RNA” (now known as messenger or mRNA) and bring specific amino acids to the “microsomal particles” (now known as ribosomes) to be incorporated into a growing polypeptide. It was also proposed that enzymes would be best suited to attach the different amino acids to the suitable adaptor molecules. We now know these enzymes as aminoacyl-tRNA synthetases and the resulting adaptor molecules charged with amino acids as aminoacyl tRNAs.

Crick further introduced the “coding problem” and suggested that code consists of triplets of nucleotides (codons) that are non-overlapping [6]. Several experiments by various groups followed and included mutagenesis of bacteriophages [7] and using cell extracts to translate, e.g., poly-U mRNA, which resulted in proteins consisting entirely of phenylalanine [8]. The anticodon tRNA part that recognizes the codon in the mRNA needs to be single-stranded and complementary to the codon [8]. The ribosome requires GTP to attach phenylalanine to the growing poly-phenylalanine polypeptide [8]. Subsequently, using mRNAs consisting of other codons, all 20 proteinogenic amino acids were identified. The findings about the genetic code and translation were summarized in 1966 [9]. The genetic code is a dictionary in which each codon encodes a unique amino acid among the twenty proteinogenic amino acids, but most amino acids can be encoded by more than one codon. This redundancy or degeneracy of the genetic code is the result of the simple mathematical fact that codons of two nucleotides (4 × 4 = 16 possible combinations) would not suffice to encode all twenty amino acids plus at least one Stop codon, but codons of three nucleotides (4 × 4 × 4 = 64 possible combinations) exceed the 21 possibilities needed. Consequently, amino acids are encoded by between one (tryptophan and methionine) and six (serine, leucine, and arginine) different codons, and there are three Stop codons. Figure 1 shows the currently accepted genetic code, which applies to most organisms.

The goal of this review is to give an overview of the nature and significance of synonymous mutations identified in different organisms, as well as some cell-free experiments with a focus on molecular mechanisms.

## 2. Definition of Single-Nucleotide Mutations

Mutations are generally changes in nucleic acids induced by various factors, such as physical or radiation damage or errors during replication, recombination, meiosis, or mitosis. They can affect entire chromosomes (omissions or duplications), large or small portions of DNA (insertions, deletions, or inversions), or they can be limited to just a single-nucleotide change. In this review, the focus is on single-nucleotide changes. To differentiate these variations from the typical population, e.g., a human reference genome, they are often referred to as single-nucleotide polymorphisms (SNPs) if they occur with a relatively high frequency of at least 1% in the population.

### 2.1. Single-Nucleotide Mutations Outside Coding Sequences

Coding regions or coding sequences (CDSs) are nucleotide sequences that encode proteins. Single-nucleotide changes outside coding sequences can have various effects ranging from none to increasing or decreasing expression levels of genes if they occur in the promoter or other regulatory regions. Such mutations can affect transcription themselves by increasing or decreasing the affinity to transcription factors, such as activators, repressors, or RNA polymerase itself. Alternatively, they can affect modifications to DNA, which in turn can affect transcription and other DNA processing events. Such modifications are often referred to as epigenetic modifications. Probably the best understood such modification is methylation. DNA methyl transferases (or methylases) recognize specific nucleotide sequences and methylate specific nucleotides in those sequences at specific sites. Single-nucleotide mutations can either create or remove such sites.

In prokaryotes, for example, methylation of the amino group in position 6 of adenine of the 5′-GATC-3′ restriction site can protect the own DNA against degradation, while foreign unmethylated DNA, for example, from phages, can be degraded, providing a protection mechanism for bacteria [10]. As might be expected, some phages have far fewer 5′-GATC-3′ sites than would be expected statistically, and some have their own DNA methylase, seemingly countering the bacterial defense mechanism [11]. Mutation of phage DNA could also remove 5′-GATC-3′ sites. Methylation of the parent strand, but not the daughter strand, during replication can indicate which strand needs to be repaired if replication has introduced an incorrect nucleotide [12]. Finally, varying methylation patterns in promoter regions can regulate the expression of genes in response to environmental factors [10].

In eukaryotes, methylation of C5 of cytosine in 5′-CG-3′, often referred to as CpG sites (p indicating the phosphodiester bond connecting the two nucleotides) or CpG islands when this dinucleotide appears in groups, often in promoter regions, is used to regulate gene expression. Methylated CpGs are thought to decrease affinity to transcription factors and RNA polymerase, thereby decreasing expression. In cancer cells, mutations that decrease the number of CpGs in oncogene promoter regions and mutations that create CpGs in tumor suppressor gene promoter regions have been observed [13]. In another study, mutations that lead to the removal or creation of CpGs in promoter regions of various genes involved in cancer, coronary artery disease, and diabetes mellitus were identified [14].

If single-nucleotide mutations occur at splice sites of eukaryotic genes or in exonic splicing enhancers or silencers, they can result in alternative splicing [15]. If they occur in RNA genes, they can cause mutations in rRNAs, tRNAs, etc.

### 2.2. Single-Nucleotide Mutations in Coding Sequences

If single nucleotides in CDSs are deleted or inserted, the frame of the coding sequence is shifted, which typically results in truncated or otherwise faulty proteins. This review focuses on single-nucleotide changes that keep the coding sequence in frame. Among these, there are four possible outcomes, which have been given different names in the past (Table 1).

Single-nucleotide changes in DNA either change the base in a nucleotide from one pyrimidine to the other pyrimidine (cytosine, C, to thymine, T, or vice versa) or from one purine to the other purine (adenine, A, to guanine, G, or vice versa), called a transition, or from a pyrimidine to a purine or vice versa, called a transversion. The one-letter code is used for the nitrogenous bases or their 2′-deoxyribose-5′-monophosphate derivatives called nucleotides. RNA deviates from DNA by having ribose rather than 2′-deoxyribose in the nucleic acid backbone and by having the pyrimidine thymine (T) replaced by uridine (U). In nucleotide mutations, nucleotides will be written in lower case to distinguish them from mutations of amino acids in upper-case one-letter code. If a nucleotide mutation results in a codon coding for a different amino acid, it is called a nonsynonymous mutation (NM). If it results in a synonymous codon, often by changing only the third nucleotide or, in some cases, only the first nucleotide, it is called a synonymous mutation (SM).

The term SM is relative and depends on what gene is used as a reference. In many cases, this will be a gene or genome that has been deposited in a reference database, such as the GRCH38p.14 human genome assembly deposited in the Ensembl database (current release 111 [16]).

It should be appreciated that in-frame single-nucleotide changes outside CDSs can become SMs if they are part of a functional gene and moved under an active promoter or if an intron becomes part of an exon by mutations at splice sites. Vice versa, mutations in CDSs can cease to be SMs or NMs if a gene ceases to be expressed due to the movement of the gene or its promoter or if an exon is not expressed because of alternative splicing.

## 3. Impact of Synonymous Mutations at the Molecular Level

### 3.1. Transcription Efficiency

The effect of SMs on transcription efficiency is not that clear. The DNA double helix structure is generally quite homogeneous except when modified by methylation or packaged with histones in eucaryotic cells [17]. The GC content of DNA could affect the ease with which the two strands of the double helix are separated because GC Watson–Crick base pairs form three hydrogen bonds versus two AT base pairs. However, a single SM will have a minimal effect on the energy required to separate the strands. If there are several SMs, it is likely they will cancel out their effects of increasing or decreasing GC content.

Single-nucleotide mutations outside CDSs can lead to the effects described above in Section 2.1, which typically occur in promoter regions. However, effects of codon usage in CDSs on transcription efficiency have also been described [18]. GC-rich DNA seems to be transcribed more efficiently than AT-rich DNA [19,20]. Such effects could be mediated by certain DNA methylation patterns, but one study found that rather histone methylation was related to the suppression of expression of genes with non-optimal codons [21]. Gene regulatory elements can also be located within CDSs.

Transcription elongation can be interrupted by pauses in both prokaryotes [22] and eukaryotes [23]. In bacteria, pauses can occur at specific nucleotide sequences called elemental pause signals (consensus sequence 5′-CATAGTTG-3′) [22]. From here, DNA polymerase can backtrack and initiate transcription-coupled DNA repair. Pauses can also be caused by hairpin loops (or other secondary structures) of the nascent mRNA or the binding or regulator proteins, including ribosomes [22]. In humans, elongation pauses occur, among other things, at DNA crosslinked by anti-cancer chemotherapeutic crosslinking or alkylating agents [23]. This is an active field of research where much remains unknown. It seems reasonable to assume that many of these processes could be affected by SMs.

### 3.2. Translation Efficiency

SMs can affect the molecular processes of translation described in the Introduction in various ways. One phenomenon often invoked is codon usage bias (CUB) or simply codon bias [24,25,26]. There can be several synonymous codons for the same amino acid, but each codon has its own cognate tRNA. Typically, there is one aminoacyl-tRNA synthetase per amino acid (except lysine, which has two) that attaches it to the 3′ end of its various tRNAs [27]. Depending on the abundance of different aminoacyl tRNAs, some synonymous codons are believed to lead to more efficient translation than others. This idea is supported by statistical analysis of codon usage between different organisms and even between different regions of genomes of the same organism. While omnipresent, CUB seems to be more prevalent in unicellular, fast-growing organisms [26].

Another way in which nucleotide sequence can affect translation efficiency is via the formation of inter- and intra-RNA secondary structures. A study by Wen et al. measured the time course of the *Escherichia coli* ribosome movement along an mRNA at the single-molecule level [28]. They found that the ribosome moves by about three nucleotides every 80 ms and that most of each step consists of a pause, with only a small portion (~10%) of the time dedicated to the actual translocation step. In addition, they showed that the ribosome has RNA helicase activity, and it unwinds the mRNA during translation. Using sequential GAG codons for glutamate residues resulted in an internal (within the CDS) Shine–Dalgarno sequence AGGAG [29] in the mRNA that arrested the ribosome at that site while using synonymous GAA codons removed the Shine–Dalgarno sequence and the arrest [28]. Consistent with this observation, Li et al. observed a stalling effect of internal Shine–Dalgarno sequences in *E. coli* [30]. A related study found that a Shine–Dalgarno sequence upstream of the gene (outside the CDS) resulted in stronger binding of the ribosome to that mRNA in the initiation complex but that this strong interaction disappeared once the ribosome started translation [31]. These results suggest a stalling effect by strong base pairing between the mRNA and the 3′ end of the 16S rRNA where the Shine–Dalgarno sequence complement is located. Apart from intermolecular mRNA-rRNA interaction, base pairing can also occur intramolecularly (within the mRNA), at its extreme, resulting in long hairpin loops that typically adopt a helical structure. These structures can be resolved by the ribosome but may require more energy and time to resolve than single-stranded stretches of mRNA [28].

## 4. Synonymous Mutations in Prokaryotes

Many studies on the impact of mutations on translation efficiency were carried out in *E. coli* [32,33,34,35]. The decreased expression of the endogenous *lamB-lacZ* hybrid gene was investigated in a series of studies on mutations close to the start codon and attributed to the formation of a stable hairpin loop that would make the Shine–Dalgarno sequence [29] inaccessible to the ribosome [34,35]. While in this case neither of the mutations involved was synonymous (one was outside the CDS, and one was an NM), the way in which they affected mRNA secondary structure and stability is universal and should also be valid for SMs.

In agreement with this scenario, a study of the expression of different variants of the heterologous human interferon (IFN)-γ gene with mutations 3′ (downstream) of the initiation codon revealed that SMs that resulted in more stable hairpin structures (as determined by model calculations [36]) led to decreased expression levels and decreased IFN-γ activity [32]. The authors also determined the ideal spacing between the Shine–Dalgarno sequence AGGA and the start codon to be between 8 and 11 nucleotides [32], consistent with another study [37].

Kudla et al. expressed a library of heterologous green fluorescent protein (GFP) variant genes with random SMs in *E. coli* to obtain a better understanding of how they affect expression [33]. Expression levels between 154 variants varied up to 250-fold. No correlation was found between CUB and expression levels. However, the stability of secondary structures near the ribosome binding site (calculated with a web server for nucleic acid melting prediction [38]) could account for more than half of the variation in expression levels, consistent with the findings described above [32,34,35]. In contrast, there was no correlation between the folding energy of the entire transcripts and fluorescence, which is consistent with the fact that the entire transcript does not have an opportunity to fold because transcription and translation are coupled in prokaryotes [39,40,41,42,43].

TEM β-lactamases have become popular subjects to study evolution due to their ease of screening for beneficial mutants. Naturally occurring mutations and their impact on enzyme activity, sensitivity to inhibitors, and/or stability have been summarized by Palzkill [44]. Many of these mutations have also been obtained by various directed evolution approaches [45,46,47,48]. Of note, in most of these studies, SMs were identified in addition to NMs but were not studied in detail.

Zalucki et al. [49] observed that leader sequences of secreted proteins in *E. coli*, including TEM-1, are encoded by non-optimal (also referred to as rare, minor, or non-preferred) codons. Changing especially the N-terminal ones to optimal (also referred to as frequent, major, or preferred) synonymous codons decreased the expression level at 37 °C but not at 28 °C, indicating that translation of the leader sequence must begin slowly for optimal expression. Too rapid expression of proteins in bacteria can result in decreased secretion, misfolding, degradation of the misfolded protein, or the formation of inclusion bodies [37,50]. Slow translation ramps have also been described elsewhere and proposed to prevent traffic jams along the mRNA [51,52,53].

In a massive synthetic biology approach [54], Firnberg et al. mutated each nucleotide of the *bla*_TEM-1_ gene to the other three nucleotides and changed each codon to the other 63 codons. This resulted in 2583 possible single point mutations and 18,081 possible codon substitutions that should encode all possible amino acids and stop codons at each position in the amino acid sequence. They were able to experimentally characterize 98.2% of the point mutations and 83.9% of the codon substitutions, covering the entire gene (Figure 2). They found that NMs have a much bigger and mostly deleterious impact on fitness than SMs. While beneficial SMs are spread throughout the entire gene, deleterious SMs are mostly found in the 5′-terminal half of the gene, especially in the 5′-terminal portion of the signal peptide-encoding region, consistent with the previously mentioned report [49]. High-impact mutations (mostly deleterious NMs) exert their effect mostly via decreased activity rather than decreased protein levels, suggesting that, in this case, protein activity is the limiting factor rather than protein abundance [54].

Zwart et al. investigated the effect of ten individual SMs [55] that were previously obtained by random mutagenesis and screening together with 38 NMs [48], out of which 10 were randomly selected for comparison. They could not find any clear trends of CUB or mRNA stability determined in a 45-nucleotide sliding window in variant transcripts with these ten SMs in comparison to ten NMs. The location of these 20 mutations in the *bla*_TEM_ gene is illustrated in Figure 2. The authors conclude that most of the SMs exert their effect via yet poorly understood post-transcriptional mechanisms.

Faheem et al. investigated SMs located throughout natural *bla*_TEM_ variant genes [56] (Figure 2). These SMs differentiate the natural genes encoding three enzymes, *bla*_TEM-3_, *bla*_TEM-33_, and *bla*_TEM-109,_ from a reference gene, *bla*_TEM-1a_, in addition to one or more NMs. Interestingly, two of the genes without SMs exhibited lower expression levels and conferred lower antibiotic resistance levels than their counterparts with SMs. *bla*_TEM-3_ with SMs resulted in a 4.2-fold higher expression level than the same gene without SMs. CUB is an unlikely factor for the difference in this case because the original codons result in excellent (actually higher) expression of *bla*_TEM-1a_.

Another variant gene, *bla*_TEM-109_, is expressed slightly better (about 1.4-fold) when SMs are included than when they are not. In this case, one of the SMs, c18t, is within the ribosome binding site. Indeed, when determining the mRNA folding energy of a 42-nucleotide segment, including the Shine–Dalgarno sequence, the mRNA segment with the SM had a binding free energy of −3.4 kcal/mol versus −6.3 kcal/mol for the mRNA segment without the SM, suggesting that the transcript with SMs could be more accessible to the ribosome.

No significant difference in expression levels was observed between *bla*_TEM-33_ variants containing SMs or not. *bla*_TEM-33_ with SMs seems to have evolved from a synonymous variant of *bla*_TEM-1a_ called *bla*_TEM-1b_ by only one NM (Figure 2), rendering the enzyme inhibitor resistant, while the counterpart *bla*_TEM-33_ without SMs could have evolved from *bla*_TEM-1a_ by the same NM. The latter gene was reported only recently (GenBank accession code CP069666). This observation raises the possibility that many variant genes already exist but have not been isolated, yet, or will evolve in the future.

When examining the location of SMs and NMs in Figure 2, one can observe that while NMs are distributed throughout the gene, SMs are more abundant in the N-terminal portion. In combination, this evidence indicates that SMs may have a role mostly in translation initiation and/or early elongation.

## 5. Synonymous Mutations in Eukaryotes

Protein expression in eukaryotic cells differs from that in procaryotic cells in several important aspects, apart from the fact that RNA polymerase and ribosomes are different in sequence, structure, and size:
Transcription and translation are decoupled both in space and time. Transcription occurs in the nucleus and translation in the cytoplasm, either at soluble or endoplasmic reticulum-bound ribosomes.mRNA must travel from the nucleus to the cytoplasm, and before that journey, it undergoes several modifications, including 5′-capping, 3′-polyadenylation, splicing, and binding to proteins and/or ribonucleoproteins.

As a result, the fate of mRNA and the impact of SMs may be different from those in procaryotes. While in procaryotes, the nascent mRNA needs to be accessible to ribosomes to be translated while it is still being synthesized, it can be subject to degradation or other post-transcriptional modifications in eukaryotes.

### 5.1. Synonymous Mutations in Various Eukaryotic Organisms

*Saccharomyces cerevisiae* (Baker’s yeast) is one of the simplest and best-studied single-celled eukaryotes. Shen et al. recently generated several thousands of artificial variants of a haploid *Saccharomyces cerevisiae* strain by synthesizing 150 nucleotide fragments of 21 selected nonessential genes with all possible single-nucleotide changes and then replaced the native fragments with the artificial fragments using CRISP-Cas9 [58]. The goal was to investigate what the impact of SMs was in comparison to NMs. The authors observed that SMs are much more like NMs in exhibiting detrimental effects than no effect at all, hence non-neutral, and this effect was mostly attributed to lower mRNA levels. They acknowledged that their findings need to be reproduced in other, including diploid, organisms to be generalized. Once validated, they argue that some evolutionary and dating techniques using SMs as controls may have to be revised and that SMs may be more important in causing disease in humans than previously appreciated [58]. A subsequent commentary argued that there is insufficient evidence to overturn a large body of evidence that SMs in humans are predominantly neutral [59]. They argue that SMs are under much smaller purifying selection and that in databases, most mutations associated with human traits are NMs. Another commentary cited technical issues related to insufficient controls and the use of technical rather than biological replicates for statistical analyses [60].

Another recent study investigated mRNA stability in yeast mediated by optimal synonymous codons [61]. A codon stabilizing coefficient (CSC) was defined by correlating the frequency of a particular codon in mRNAs with the mRNA half-life. High-CSC mRNAs were significantly more stable than corresponding low-CSC mRNAs encoding the same protein, but this effect only emerged above an mRNA length of a few hundred nucleotides, including the 5′ untranslated region. These longer mRNAs also had a higher propensity to form polysomes (more than one ribosome bound per mRNA), which could explain this increased mRNA stability [61]. SMs could be responsible for converting more stable high-CSC mRNAs into less stable low-CSC mRNAs and vice versa.

Some reviews discuss SMs in genomes of various eukaryotic species, including *S. cerevisiae*, *Arabidopsis thaliana, Drosophila melanogaster*, *Caenorhabitis elegans*, and mammals [62,63,64]. The selection of optimal synonymous codons in yeast, *D. melanogaster, C. elegans,* and humans can be explained by CUB: genes that are highly expressed and essential mostly use codons that are complementary to abundant tRNAs [62,64]. Major codons may also reduce the energetic costs of proofreading during protein synthesis [63]. Interestingly, codons in CDSs have a higher tendency to be optimal synonymous codons than codons in neutrally evolving introns, and codons in constitutively expressed exons have a higher tendency to be optimal than those in alternatively spliced (rarely expressed) exons [65].

### 5.2. Synonymous Mutations Associated with Human Pathophysiology

Many SMs in humans (also called synonymous single-nucleotide polymorphisms or sSNPs) have been discovered due to their pathological effects in genome-wide association studies (GWAS) [15,66]. According to these reviews, more than 50 human diseases are associated with SMs. In addition, Hunt et al. summarize both computational and experimental methods that have been employed in the study of SMs [15]. Here, we will focus on a few examples where the underlying molecular mechanisms are quite well understood.

Duan et al. [67] reported that naturally the human dopamine receptor D2 (DRD2) gene has a high GC content at the third positions of its codons. One SM, c957t, decreased protein expression level, while its combination with g1101a restored the original expression level. In Chinese hamster ovary cells transfected with the wild-type or mutant DRD2 genes and with transcription arrested by the addition of actinomycin D, the mRNA expressed from the mutant gene carrying the c957u SM decayed about twice as fast as the wild-type mRNA and mutants carrying other SMs. The combination of c957u and g1101a stabilized the mRNA. Using *mfold* [68], the authors investigated if the synonymous mutations had an impact on mRNA secondary structure, which could affect stability. They found that the c957t SM changed the mRNA structure relative to wild-type mRNA, while the g1101a SM maintained a structure very similar to the wild type. Adding both SMs resulted in an mRNA structure very similar to the one of the g1101a variant and similar to the wild type. Contrary to what might have been expected, the two nucleotide pairs (c957 and g1101; and t957 and a1101) did not form direct base pairs but affected the mRNA secondary structure in different regions. In line with the observations at the beginning of this section, in eukaryotic organisms, instability of mRNA can be detrimental because the mRNA needs to endure the journey from the nucleus to the site of translation. In contrast, in prokaryotes, instability of mRNA can be beneficial if it allows the ribosome to bind to the ribosome binding site more efficiently.

Bartoszewski et al. [69] investigated a three-nucleotide deletion in the cystic fibrosis transmembrane conductance regulator (CFTR) gene, which is a common cause of cystic fibrosis. A three-nucleotide (CTT) deletion removes the first two nucleotides of the TTT F508 codon, resulting in the expression of ΔF508 CFTR, and the third nucleotide from the ATC I507 codon, resulting in a synonymous ATT codon. By mutating the T in the ATT I507 codon of the deletion mutant back to C, the authors effectively compared the impact of an SM. In silico mRNA folding using *mfold* [68] indicated that the AUC -> AUU synonymous mutation coding for I507 rather than the deletion of UUU coding for F508 caused a different secondary structure. They compared the wild-type mRNA containing AUC UUU to the deletion mutant containing AUU and saw some differences in circular dichroism spectra. They carried out cell-free expression and expression in 293F cell lines. In both cases, the protein expressed from the gene with the ATC I507 codon was expressed at higher levels than from the gene with the ATT I507 codon. Transcription levels and mRNA stability were comparable, pointing to translation as the process determining expression levels. The authors also observed that the protein expressed from the gene with the ATC I507 codon was less susceptible to endoplasmic reticulum-associated degradation than the one expressed from the gene with the ATT I507 codon. They argue that the slower expression of the protein from the latter gene could cause slower protein folding, more ubiquitination [70], and more degradation.

In a subsequent review [71], Bartoszewski et al. summarize the effect of SMs in other human diseases, such as amyotrophic lateral sclerosis (ALS), pain perception, cancer, and multi-drug resistance. They also demonstrate a method for predicting translation efficiency based on whether optimal or rare codons are used. Applied to CFTR, this allowed them to predict which parts of the protein are translated quickly (e.g., cytosolic loops and domains and extracellular loops) or slowly (e.g., α helices in the two transmembrane domains) [71]. Interestingly, it was also shown that ΔF508 CFTR expressed from a gene with the ATC I507 codon is a more functional chloride channel than the same protein expressed from a gene with the ATT I507 codon [72] and that the two proteins had different sensitivity to drugs [73]. These results suggest that in this case, the SM not only affected the translation level but also translation-associated protein folding and subsequent protein function.

Silencing via the binding of small noncoding RNAs (also known as microRNAs or miRNAs) [15] is a post-transcriptional regulation mechanism that can be affected by SMs in metazoans. Human immunity-related GTPase family M protein (IRGM) expression is usually silenced by miR-196, but this silencing effect is decreased and leads to overexpression of IRGM with an SM. Increased IRGM expression leads to deregulated xenophagy in Crohn’s disease [74]. Another example where miRNA silencing was disrupted by an SM is decreased silencing of the BCL2L12 gene, typically silenced by hsa-miR-671-5p. Increased expression of this gene triggers anti-apoptotic signaling in some patients with melanoma [75].

## 6. Conclusions

Despite an increasing body of evidence many questions about the causes, molecular mechanisms, and effects of SMs remain unanswered. There is interest in this field, and the language in the scientific community has shifted from silent mutations to synonymous mutations, recognizing the fact that the effect of SMs cannot be discounted (Figure 3).

The number of publications using the search term “silent mutation*” increased from the 1970s (shortly after the genetic code was cracked), peaked around 2000, and has been steady since then, probably in part because authors refer to the “historical” term silent mutations. The first publication obtained using the search term “synonymous mutation*” appeared in 1987, and the number of these publications has increased significantly since 2000 to about three times the number of publications using “silent mutation*” today. It is expected that this number will continue to increase, if not due to increased interest, then simply because sequence information is increasing exponentially and, with it, annotation of SMs.

The biggest controversy seems to revolve around how significant the impact of SMs is, especially for human disease [15,58,59,60,66]. It remains to be seen if the examples described here are part of a general trend or outliers where SMs just happened to have important effects. The beauty of science is that controversy is a driving force for new discoveries. The human genome was decoded more than two decades ago [76,77], but much remains unknown about how our genes determine human bodies, physiology, thoughts, and disease. As more of these determinants are deciphered, SMs will certainly be part of the equation.

## Figures and Tables

**Figure 1 biomolecules-14-00132-f001:**
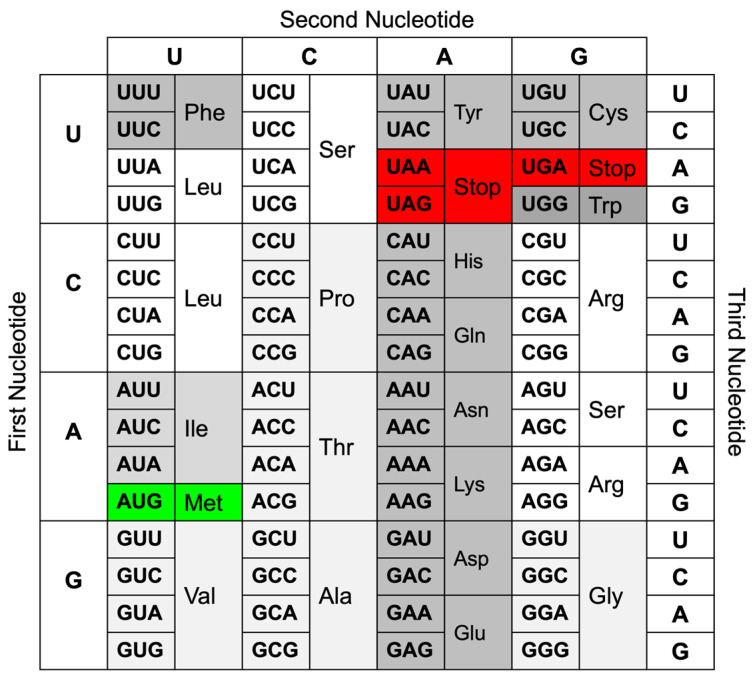
Genetic code redrawn as presented by Crick [9], except that the OCHRE AMBER, and ? codons were replaced with Stop codons. The Start codon AUG and its encoded amino acid methionine are colored in green, and the three Stop codons are colored in red. All other codons are shaded in different gray tones depending on their redundancy: the single tryptophan (Trp) codon in dark gray; pairs of synonymous codons for phenylalanine (Phe), tyrosine (Tyr), histidine (His), glutamine (Gln), asparagine (Asn), lysine (Lys), aspartate (Asp), glutamate (Glu), and cysteine (Cys) in slightly lighter gray; the three isoleucine (Ile) codons in lighter gray; quartets of synonymous codons coding for glycine (Gly), alanine (Ala), valine (Val), proline (Pro), and threonine (Thr) in the lightest gray; and sextets of synonymous codons coding for serine (Ser), leucine (Leu), and arginine (Arg) in white.

**Figure 2 biomolecules-14-00132-f002:**
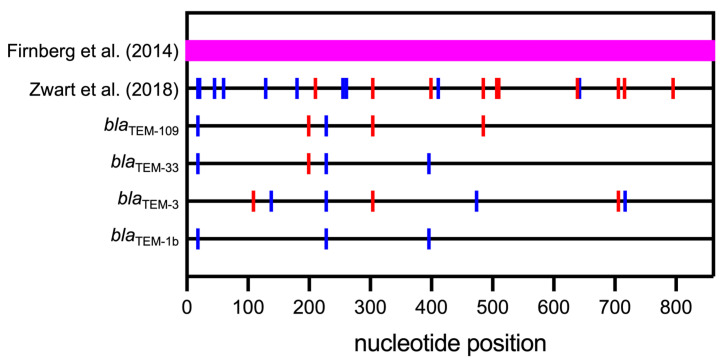
Positions of SMs (blue) and NMs (red) in the *bla*_TEM_ genes studied relative to *bla*_TEM-1a_. Firnberg covered the entire sequence (861 nucleotides) with both SMs and NMs indicated by the magenta band. Data are taken from Firnberg et al. [54], Zwart et al. [55], *bla*_TEM-3,_
*bla*_TEM-33,_ and *bla*_TEM-109_ are taken from Faheem et al. [56], and *bla*_TEM-1b_ is taken from Boyd et al. [57].

**Figure 3 biomolecules-14-00132-f003:**
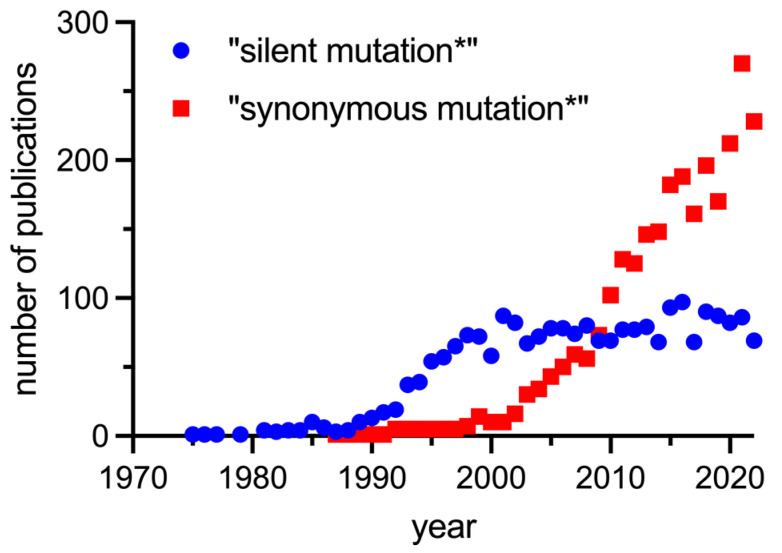
Number of publications found in PubMed using the search terms “silent mutation*” and “synonymous mutation*”.

**Table 1 biomolecules-14-00132-t001:** Types of in-frame single-nucleotide changes in CDSs.

Name Used in This Review	Outcome	Alternative Names
Synonymous mutation (SM)	No change in amino acid (AA) or Stop codon	Silent
Nonsynonymous mutation (NM)	Change from one AA to another AA	Missense
	Change from an AA to a Stop codon	Nonsense, Stop
	Change from a Stop codon to an AA	Nonstop

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
