# Peer review of "Molecular Mechanisms and the Significance of Synonymous Mutations"

_biomolecules, 2024, doi:10.3390/biom14010132_

Round 1

Reviewer 1 Report

Comments and Suggestions for Authors

The author’s review on codon optimality and synonymy covers bacteria, yeast as well as mammals. The review highlights some controversial findings, some of them may simply be due to experimental inconsistencies.

Suggested changes / extensions.

1.      Readability: The author describes some experiments in great detail, mentioning even mutant names. This is fine as long as figures are provided, which is for example the case for Fig.2. However, some experiments such as the compensation between the mutations through RNA folding, no figure is provided. The authors should either provide figure or alternatively formulate the general principles of the finding without mentioning of the mutant names. This makes the reading easier.  

2.       “3.1. Transcription efficiency. The author writes that single mutations are likely to have negligible effect due to the averaging of AT content. However, CpG islands are affected less by averaging and thus SNP may influence transcription.

https://pubmed.ncbi.nlm.nih.gov/33128106/

https://pubmed.ncbi.nlm.nih.gov/27074591/

https://pubmed.ncbi.nlm.nih.gov/35246549/

3.        The author mentions controversies but few examples are provided when the controversy is explained (e.g. the example with RNA folding). The authors should provide more general findings that explain apparent controversies. For example,  mRNAs with short coding sequences are expected to be more affected by single silent mutations because of less averaging. However, codon optimality fails to affect mRNA stability and translation in short mRNAs because the decoding of codon synonymy into mRNA stability and translation requires polysomes.

https://pubmed.ncbi.nlm.nih.gov/37756413/

Reviewer 2 Report

Comments and Suggestions for Authors

The Introduction (Section 1) is rather extensive by including a historical perspective - this could be made more concise and focused on the topic of the review.

Section 2 is also rather long and more like a molecular biology textbook, rather than a review in a scientific journal. But then there is a very sudden, highly specialized example in Lines 126-136 that seems to be out of place with the style of the previous paragraph. If specific examples are mentioned, there should be more than this one (otherwise this looks very out of balance by focusing attention), maybe also using some human gene.

In Section 3, the effects on transcription efficiency are brushed away to easily (Lines 138-146). The function of transcription pause- and arrest sites could very well be changed by SMs, or binding motifs for gene-specific transcription factors enhanced or diminished. The formation of specific secondary structures of the transcript could also affect the elongation properties of RNA polymerases. While I concur with the general direction of the argument presented (that is unlikely to cause a major transcriptional effect, at least the possibilities of how it might could be discussed briefly).

Again, do we need details of amino acetylation chemistry in Lines 153-156?

The end of Section 3 and beginning of Section 4 appear to be thematically related. Should these be really in two adjacent sections, or could the topic be combined in a more consistent manner within just one section?

Line 255-258: The interpretations offered here are very indistinct - these should be made clearer.

The entire section (Line 184 - 321) contains a lot of unnecessary details that should be summarized more effectively and only key conclusions should be presented - this is a general review and I found most of the results presented only moderately interesting because they are far too specialized (and often do not provide particularly conclusive insights either!)

In Section 5, a lot of information that would be relevant to this review is only referred to very briefly (Lines 55-57). While I accept that these topics have been covered in other reviews, at least some of the key points and conclusions should be reiterated here to make the current review more authoritative.

In summary, the current version requires more balance between the very general background provided in some sections (especially the Introduction) and very specialized information provided in other sections (especially Section 4). This creates a situation where neither inexperienced beginners nor readers with more advanced knowledge of the field will find this review particularly enjoyable to read. I urge the author to re-balance the sections and consider what information is most relevant (and cutting some parts) before going ahead with publication.

Reviewer 3 Report

Comments and Suggestions for Authors

In this review, the author discusses about synonymous mutations (SM) which do not alter amino acid sequence of proteins. More specifically, the review introduces how SM may influence the expression of a protein without exchanging any amino acid as well as the underlying mechanism. Notably, mutations that cause alternative splicing or localize to non-coding regions were excluded.

The author starts with an introduction to some history of the “coding problem” and defines the scope of “synonymous mutations” that will be discussed. Then the author points out that SM mainly affects translation efficiencies and gave a few potential mechanisms. In the main body of the text, the author discusses about SM in prokaryotes and eukaryotes. The logic and writing of this review are clear and easy to follow. This review summarizes what is already known about SM and covers a good selection of published works.

This referee finds the review suitable for publication in Biomolecules with minor revisions.

Specific comments:

1. This referee finds the introduction to the establishment of some fundamental concepts like the genetic codons and central dogma very fascinating and educational. However, it is not well aligned with the topic of this review and some trimming seems necessary.

2. I am not an expert but what do we know about SM in viruses? It seems to be an interesting topic to include.

3. Is it possible that some SMs create more potential for non-synonymous mutations (more AA possibilities) and as a result are favorable? This is just a curious question by this referee and may not go into the review.

Round 2

Reviewer 1 Report

Comments and Suggestions for Authors

The author addressed all my comments.

Reviewer 2 Report

Comments and Suggestions for Authors

The author has followed my feedback for improving the readability of the manuscript.

Comments on the Quality of English Language

No problems.